# Antibiotic Susceptibility Profile of *Pseudomonas aeruginosa* Canine Isolates from a Multicentric Study in Romania

**DOI:** 10.3390/antibiotics10070846

**Published:** 2021-07-12

**Authors:** János Dégi, Oana-Alexandra Moțco, Diana Maria Dégi, Tiana Suici, Mihai Mareș, Kalman Imre, Romeo Teodor Cristina

**Affiliations:** 1Department of Infectious Diseases and Preventive Medicine, Faculty of Veterinary Medicine, Banat’s University of Agricultural Sciences and Veterinary Medicine “King Michael I of Romania” from Timisoara, 300645 Timisoara, Romania; janosdegi@usab-tm.ro (J.D.); kalmanimre@usab-tm.ro (K.I.); 2Dialab Solutions, 011607 Bucharest, Romania; oana.motco@aadialab.ro; 3Department of Pharmacology and Pharmacy, Faculty of Veterinary Medicine, Banat’s University of Agricultural Sciences and Veterinary Medicine “King Michael I of Romania” from Timisoara, 300645 Timisoara, Romania; diana.maria.degi@gmail.com; 4Department of Dermatology, Faculty of Veterinary Medicine, Banat’s University of Agricultural Sciences and Veterinary Medicine “King Michael I of Romania” from Timisoara, 300645 Timisoara, Romania; sujic.tijana@yahoo.com; 5Department of Public Health, Faculty of Veterinary Medicine, “Ion Ionescu de la Brad” University of Life Sciences, 700489 Iasi, Romania; mmares@uaiasi.ro

**Keywords:** *Pseudomonas*, antibiotic resistance, dog, infection, skin, otitis externa, perianal abscess

## Abstract

Treating infections caused by *Pseudomonas aeruginosa* is increasingly difficult due to high antibiotic resistance, materialized through the presence of multiple resistance strains, as well as due to rapid development of resistance throughout treatment. The present survey was conducted to investigate the antibiotic susceptibility profile of *Pseudomonas aeruginosa* pathogens in two University Veterinary hospitals from different geographical regions of Romania (i.e., Southwest Timișoara county and Northeast Iași county) involved in superficial canine infections. A total of 142 swab specimens were collected from dogs with superficial infections (superficial skin infections, otitis externa, and perianal abscess) to assess the presence of *Pseudomonas* *aeruginosa*, based on phenotypic and molecular characterization. According to their confirmed morphological and molecular features, 58 samples (40.84%; 58/142) were positive for *Pseudomonas* *aeruginosa* (according to their confirmed morphological and molecular features). Antibiotic susceptibility testing for 12 antibiotics was conducted using the Kirby–Bauer disc diffusion method. Drug resistance was observed in the case of all tested antibiotics. The susceptibility rate of *P. aeruginosa* strains that were tested in this study was in the following order: ceftazidime (53.44%; 31/58), followed by aztreonam (51.72%; 30/58), amikacin (44.82%; 26/58), azithromycin (41.37%; 24/58), gentamicin (37.93%; 22/58), cefepime (36.20%; 21/58), meropenem (25.86%; 13/58), piperacillin-tazobactam (25.86%; 13/58), imipenem (22.41%; 13/158), ciprofloxacin (17.24%; 10/58), tobramycin (8.62; 5/58), and polymyxin B (1.72; 1/58). The results highlight the importance of antibiotic susceptibility testing in *Pseudomonas* *aeruginosa* isolates from dogs with superficial infections to use an adequate treatment plan to manage the skin condition and other pathologies (otitis externa and perianal abscesses).

## 1. Introduction

*Pseudomonas aeruginosa* is an essential pathogen to both humans and animals, but it is rarely involved in primary diseases. In humans, *P. aeruginosa* is an important opportunistic, nosocomial pathogen, mainly present in hospital-acquired pneumonia cases in immunocompromised patients. In animals, and especially in dogs, it has been considered the distinct cause of infections such as otitis externa, superficial skin infections, chronic deep pyoderma, perianal abscesses, and wound/urinary tract infections [1,2].

The prevalence of *P. aeruginosa* infections is 11.5% in Europe and 17% in developing countries [1]. Among the resistant bacteria, *P. aeruginosa* expresses resistant to antibiotics that can be either acquired (plasmids, transposons) or natural. This resistance generally favors the involvement of *P. aeruginosa* in nosocomial infections, food poisoning, and biofilm formation, the latter giving *P. aeruginosa* high colonization potential, the capacity to spoil foodstuffs, and resistance to antiseptics, disinfectants, and antibiotics [2,3].

Pyoderma is defined as an inflammatory skin condition of bacterial origin, most commonly characterized through a purulent aspect. The skin is a complex ecosystem hosting various microorganisms such as bacteria, yeasts, and parasites. Animals with a high skin humidity index are the perfect environment where such microorganisms thrive [1,2,3]. *P. aeruginosa* plays an essential role in the pathogenesis of canine pyoderma. Isolation and identification of microbial agents involved in skin disorders of dogs are a fundamental starting point for diagnosis and for initiating a suitable treatment [2,4]. Treatment of *P. aeruginosa* infections is complex because of its high intrinsic resistance to many commonly used antibiotics. Therefore, the choice of antibiotics that can be used to treat *P. aeruginosa* infections is scarce, especially in veterinary medicine [1,2,3,4].

Bacteria are found on the skin surface, in the superficial part called stratum corneum, but they are absent in the external area of the hair follicles and up to the sebaceous gland. Several bacterial species can coexist in harmony without causing any damage to the skin; however, instability can occur at any point [4,5,6,7]. The skin microbiota consists of resident bacteria and occasionally transient bacteria. The resident bacteria can multiply on the skin surface and in the hair follicles, but their presence is non-pathogenic. Thus, pyoderma is most often a secondary rather than a primary disease. Canine pyoderma only occurs if there is an association between pathogenic bacteria and the factors that allow their proliferation and penetration through the skin [8,9].

*Pseudomonas* infections in dogs with skin disorders have been reported to be around 11–13% [4,10,11,12]. In chronic, suppurative diseases, *Pseudomonas aeruginosa* is the dominant bacterial species, isolated alone or in association with other microorganisms (especially *Proteus mirabilis* and *Staphylococcus* spp. [1,8,13,14]. Due to its high resistance to antibiotics, treating infections caused by *Pseudomonas aeruginosa* is increasingly difficult. The known risk factors for selecting resistant and multidrug-resistant strains are excessive drug use and inappropriate dosage regimens without previous antibiotic susceptibility testing [15]. Therefore, considering all the reasons mentioned above, we value the determination of antibiotic susceptibility of *Pseudomonas aeruginosa* strains involved in pet infections to be of the utmost importance [3,16,17].

All these considered, the purpose of the study was to determine the antibiotic susceptibility of *P. aeruginosa* strains isolated from canine skin superficial infections cases in two clinical settings from Romania (i.e., University Veterinary Hospital-Timișoara County and University Veterinary Hospital-Iași County).

## 2. Results and Discussion

A total of 58 (40.86%) bacterial isolates showing typical characteristics of the *Pseudomonas aeruginosa* species were isolated from superficial infections (including superficial pyoderma, otitis externa, and perianal abscess) lesions. The distribution of positive isolates among 142 canine patients is presented in Table 1.

All 58 isolates were confirmed positive for *Pseudomonas aeruginosa* by molecular methods. In the present study, the percentage of samples collected from dogs with superficial infections positive for *P. aeruginosa* was 40.86% (58/142; *n* = 58). The susceptibility rate of *P. aeruginosa* strains that were tested was in the following order: ceftazidime (53.44%; *n* = 31), aztreonam (51.72%; *n* = 30), amikacin (44.82%; *n* = 26), azithromycin (41.37%; *n* = 24), gentamicin (37.93%; *n* = 22), and cefepime (36.20%; *n* = 21). Other similar studies reported that gentamicin could have an increased efficacy against *Pseudomonas* strains of animal origin, regardless of the animal species or isolation site [4,8]. Good susceptibility rates were also communicated in other studies for polymyxin B, a common component of the topical preparations [3,9,18].

The results of the antimicrobial susceptibility tests performed on 58 *Pseudomonas aeruginosa* strains isolated from superficial canine infections are presented in Table 2, and the antibiotic resistance/susceptibility pattern of the multidrug-resistant *Pseudomonas aeruginosa* strains (*n =* 18) is shown in Table 3.

Eighteen out of 58 *Pseudomonas aeruginosa* isolates showed resistance to at least 10 of the 12 antibiotics tested (Table 3). All 58 isolates from superficial canine infections were resistant to multiple antimicrobial classes, including synthetic antimicrobial agents that are not commonly used in canine infections management but mainly used in human medicine.

Significant variability of susceptibility profiles was observed among these isolates—the situation for each isolate belonging to a different type of disease is detailed in Table 4, Table 5 and Table 6. The higher resistance rates were encountered for polymyxin B (98.27%), tobramycin (91.37%), and ciprofloxacin (82.75%), as shown in Table 2.

Bacterial skin conditions are common in dogs, and their empirical treatment is a general therapeutic approach to reducing clinical evolution. Still, in severe forms, antibiotic susceptibility testing should be considered imperative.

*Pseudomonas* isolates in canine pyodermatitis have been confirmed in numerous other studies carried out in several geographic regions [19,20]. In agreement with our findings regarding the presence of multidrug-resistant *Pseudomonas* in canine pyodermatitis lesions, multiple data are reported in previous surveys conducted by Wildemuth et al. [21], based on comparing the susceptibility of *Pseudomonas* spp. isolates from skin and ear disorders, towards enrofloxacin, marbofloxacin, and ciprofloxacin. Pathological exudates were obtained from dogs examined within the dermatology wards of veterinary hospitals. The susceptibility rate of the isolates from ear infections was 46.90% to enrofloxacin, 66.70% to marbofloxacin, and 75.0% to ciprofloxacin. The isolates from the skin showed the following susceptibility pattern: 76.20% to enrofloxacin, 81.0% to marbofloxacin, and 80.0% to ciprofloxacin [21].

In a study conducted by Hillier et al. [8], based on the examinations of 20 dogs with different skin conditions, the authors reported that 33.0% of the cases were cases of pyoderma caused by *Pseudomonas*, susceptible to florfenicol, which was also the treatment option in their study [8].

In another study, Morris [3] reported mixed results regarding the in vitro susceptibility of *P. aeruginosa* strains isolated from the ear to fluoroquinolones. The differences found in the diffusion susceptibility tests were as follows: 58.0% were enrofloxacin-susceptible strains, and 96.0% were marbofloxacin-susceptible strains, out of a total of 26 strains of *Pseudomonas* spp. (of which 25 were *P. aeruginosa* strains), isolated from the ear [3].

*P. aeruginosa* has both intrinsic as well as acquired mechanisms of resistance against β-lactams.

Intrinsic include influx pumps, as well as several β-lactamases. Specifically, *P. aeruginosa* has chromosomally encoded *AmpC* β-lactamases and extended-spectrum-β-lactamases (ESBLs) [22].

Acquired β-lactam resistance mechanisms in *P. aeruginosa* include mutations in porins, such as deficiency of the *OprD* porin, which leads to high-level resistance to imipenem and other carbapenems; overexpression of hydrolyzing enzymes such as *AmpC*; over-expression of efflux pumps such as MexCD–OprJ, which reduces susceptibility to carbapenems; modification of PBPs, which reduces susceptibility to several β-lactams; and finally, acquisition of other β-lactamases, such as Class B carbapenemases [22,23,24].

*AmpC* is cephalosporinase, which can hydrolyze most penicillins, early-generation cephalosporins, and combinations of β-lactam and β-lactamase inhibitors. The results from this study support the importance of recommending veterinarians to conduct AST to guide antimicrobial treatment in dogs. Furthermore, the high resistance frequency detected towards clinically important antimicrobials in *P. aeruginosa* isolated from dogs creates important therapeutic challenges and points to the need for promoting programs for the prudent use of antibiotics in Romania. Antipseudomonal cephalosporins such as ceftazidime or cefepime, while still susceptible to *AmpC*, are weak inducers of its expression. However, the prolonged administration of anti-pseudomonal β-lactams can lead to the selection of *P. aeruginosa* isolates that overproduce *AmpC* and subsequently treatment failure [24].

## 3. Materials and Methods

### 3.1. Animals Handling

A total of 142 dogs with characteristic clinical signs of skin superficial infections, otitis externa, and perianal abscesses were presented at the University Veterinary Hospitals of the Faculty of Veterinary Medicine of Timisoara (FVMT), Western Romania, and Faculty of Veterinary Medicine of Iasi, Eastern Romania, from 1 January to 30 June 2019.

The dogs in this study were selected and examined during routine veterinary visits as part of a diagnostic workup. No treatment decisions were made based on the results of the clinical examination. All methods were conducted in accordance with relevant guidelines and regulations. Because the samples were intended for the diagnosis, the collection protocol was carried out with the consent of animal owners, according to the code of the Romanian Veterinary College (protocol numbers 34/1.12.2012) and the proper procedures of the University Veterinary Clinics of the Faculty of Veterinary Medicine Timișoara and Iași.

Exclusion criteria were: lactating and pregnant bitches, absence of skin lesions, immunosuppressed subjects.

Inclusion criteria were: the animals included in the study were dogs, aged five months to 14 years, of both sexes (69 males and 73 females), belonging to 21 different breeds.

The main criteria for inclusion in the study were:

a. The absence of any antibiotic treatment before clinical presentation;

b. Presence of main clinical signs of superficial skin infections: excessive scaling, follicular papules, epidermal collarettes, serous crusts, erythema, serohaemorrhagic or purulent exudates, ear discharge or desquamation, local pain according to criteria from Table 7.

The first assessment of the patients included the following steps: clinical examination, scoring the extent of superficial infections, collecting of the exudates. Exudates from the skin surface, external ear canal, and perianal glands were collected using sterile cotton swabs (Firatmed FT20-60 Transport Swab, Sincan Ankara, Turkey) dipped in a sterile saline solution before sampling. Samples were maintained in Amies transport medium (M40 Transystem™; Copan, Brescia, Italy) until laboratory processing.

All superficial skin purulent lesions and perianal zones were sanitized using sterile saline solution before sample collection. This procedure led to a decrease in microbial contamination. Fresh exudates were easily obtained by applying light pressure on the lesion areas (skin and perianal region).

### 3.2. Bacteriological Examination

The microbiological examinations were performed in the Bacteriology Laboratory of the Infectious Diseases and Preventive Medicine Department, FVMT, and Laboratory of Microbiology, FMVI. Analyses were completed within three hours of sampling or, in some cases, after 24–48 h. During the waiting period, the clinical samples were refrigerated in particular storage environments. The collected samples were inoculated onto CHROMagar™ *Pseudomonas* (CHROMagar, Paris, France) and Columbia Agar with 5.0% sheep blood (Becton Dickinson, GmbH, Heidelberg, Germany). The plates were incubated for 18–24 h at 37 °C, in aerobic conditions. The presumptive identification of isolates was based on the cultural, morphological, and biochemical characters.

### 3.3. Molecular Tests

The final species identification was performed by PCR using a previously described method [23]. All *Pseudomonas aeruginosa* isolates give an amplified fragment of 956 bp (Figure 1).

#### Extraction of Template DNA and Polymerase Chain Reaction (PCR)

One milliliter of bacteria grown in 10 milliliters of Cetrimide broth, at 37 °C, in an aerobic atmosphere for 24 h, was dispensed aseptically in a microcentrifuge tube. Bacterial genomic DNA was extracted using the Pure Link™ Genomic Lysis/Binding Buffer (Thermo Fisher Scientific, Basingstoke, UK) boiling method, with a freshly prepared proteinase K solution (10 mg/mL). The DNA quantity and quality were determined using a NanoDrop ND-1000 spectrophotometer (NanoDrop^®^ Technologies, Thermo Fisher Scientific, Basingstoke, UK) by measuring the absorbance at 260 nm.

PCR was done using a specific primer for identifying *Pseudomonas aeruginosa* species, PA-SS-F GGGGGATCTTCGGACCTCA and PA-SS-R TCCTTAGAGTGCCCACCCG 1124–1144 as previously described [25].

The enhanced PCR conditions consisted of an initial denaturation at 95 °C for 5 min followed by 32 cycles of denaturation at 95 °C for 1 min, annealing at 55 °C for 1 min, extension at 72 °C for 1 min, and a final extension at 72 °C for 10 min, using the thermocycler My Cycler (BioRad^®^, Dubai, United Arab Emirates). The amplified products were analyzed for their size by electrophoresis on 2.5% agarose gel, stained with ethidium bromide, and visualized under UV light using a gel documentation system (UV transilluminator–2035-2, Bio Olympics Ltd., Thousand Oaks, CA, USA). The type strain *Pseudomonas aeruginosa* ATCC 27853™ was used as the positive control.

### 3.4. Antimicrobial Susceptibility Tests

A standard method for determining antibiotic susceptibility, especially in small laboratories and veterinary practices, is the agar diffusion test (Kirby–Bauer disc diffusion method). This test was performed using Müller–Hinton Agar (Becton Dickinson GmbH, Heidelberg, Germany).

The following antibiotic discs (Bio-Rad, Marnes-la-Coquette, France) were tested: gentamicin (GM/10 µg; disk diffusion clinical breakpoints: S ≥ 15; I = 13–14; R ≤ 12); ciprofloxacin (CIP/5 µg; disk diffusion clinical breakpoints: S ≥ 21; I = 16–20; R ≤ 15); imipenem (IPM/10 µg; disk diffusion clinical breakpoints: S ≥ 19; I = 16–18; R ≤ 15, according to CLSI M100-S18); meropenem (MEM/10 µg; disk diffusion clinical breakpoints: S ≥ 19; I = 16–18; R ≤ 15, according to CLSI M100-S28); piperacillin-tazobactam (TZP 110/100 + 10 µg; disk diffusion clinical breakpoints: S ≥ 21; I = 15–20; R ≤ 14); ceftazidime (CAZ/30 µg; disk diffusion clinical breakpoints: S ≥ 18; I = 15–17; R ≤ 14); cefepime (FEP/30 µg; disk diffusion clinical breakpoints: S ≥ 18; I = 15–17; R ≤ 14); aztreonam (ATM/30 µg; disk diffusion clinical breakpoints: S ≥ 22; I = 16–21; R ≤ 15); azithromycin (AZM/15 µg; disk diffusion clinical breakpoints: S ≥ 19; I = 15–18; R ≤ 14, according to *Enterobacteriaceae* reference); amikacin (AN/30 µg; disk diffusion clinical breakpoints: S ≥ 17; I = 15–16; R ≤ 14); tobramycin (TM/10 µg; disk diffusion clinical breakpoints: S ≥ 15; I = 13–14; R ≤ 12); and polymyxin B (PB/50 µg-300UI; disk diffusion clinical breakpoints: S ≥ 12; I = 0; R ≤ 11).

Antibiotics were chosen based on their importance to veterinary and human medicine, and antibiotic susceptibility was evaluated, considering the inhibition zone diameter [26]. According to these criteria, *Pseudomonas aeruginosa* strains were classified as sensitive, intermediate, or resistant [27].

The control strain *Pseudomonas aeruginosa* ATCC 27853™ was also used in this study [8].

## 4. Conclusions

The use of antibiotic susceptibility tests before choosing the therapeutic protocol is of specific importance. The emergence of multidrug-resistant *Pseudomonas aeruginosa* strains, especially those resistant to not commonly and effectively used antibiotics in dogs (but used in human medicine), is also an occurring event that might indicate the overuse of these antimicrobial agents or is suggestive of human-to-dog transfer. As a result, when choosing antibiotics, veterinary surgeons must consider the following aspects of site-specific prevalence of antibiotic-resistant *Pseudomonas aeruginosa* in dogs:

(a) The therapeutic confrontation that occurs due to these bacteria;

(b) The need to constantly conduct AST to determine the suitable antibiotic;

(c) The significance of circumventing the use of critically important antibiotics that are not approved for veterinary use;

(d) The choice of first topic therapeutic approaches for skin and ear infections besides antibiotic use to minimize the need to prescribe human use antibiotics.

Generally, to perfect more suitable control strategies for superficial canine infections, performing microbiological exams and continuously keeping updated on the susceptibility profile of isolated strains are practices that are strongly recommended.

## Figures and Tables

**Figure 1 antibiotics-10-00846-f001:**
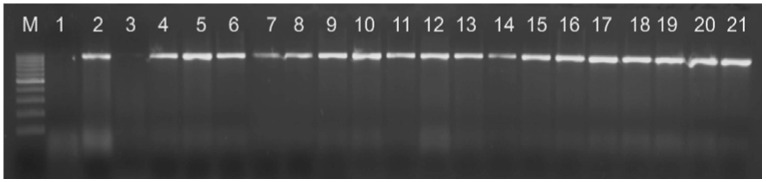
PCR amplification using *Pseudomonas* genus-specific primers, M: marker. Legend: line 1—negative control, line 2—positive control, lines 3–21: clinical isolates.

**Table 1 antibiotics-10-00846-t001:** Distribution of *Pseudomonas aeruginosa* isolates from dogs with superficial infections.

Type of Disease	*Pseudomonas aeruginosa* Distribution	
	No. of Collected Samples (*n*)	No. Positive Samples (%)
Superficial skin infections	56	19 (33.93)
Otitis externa	48	15 (31.25)
Perianal abscess	38	24 (63.15)

**Table 2 antibiotics-10-00846-t002:** Antibiotic susceptibility profile of *Pseudomonas aeruginosa* strains isolated from superficial canine infections.

Antibiotics	Antibiotic Susceptibility Test Results	*Pseudomonas aeruginosa* Strains Isolated from Canine Superficial Skin Infections (*n =* 58)	
		No.	%
Gentamicin (GM/10 µg)	S	22	37.93
	R	36	62.06
Ciprofloxacin (CIP/5 µg)	S	10	17.24
	R	48	82.75
Imipenem (IPM/10 µg)	S	13	22.41
	R	45	77.58
Meropenem (MEM/10 µg)	S	15	25.86
	R	43	74.13
Piperacillin-tazobactam (TZP 110/100 + 10 µg)	S	15	25.86
	R	43	74.13
Ceftazidime (CAZ/30 µg)	S	31	53.44
	R	27	46.55
Cefepime (FEP/30 µg)	S	21	36.20
	R	37	63.79
Aztreonam (ATM/30 µg)	S	30	51.72
	R	28	48.27
Azithromycin (AZM/15 µg)	S	24	41.37
	R	34	58.62
Amikacin (AN 30 µg)	S	26	44.82
	R	32	55.17
Tobramycin (TM/10 µg)	S	5	8.62
	R	53	91.37
Polymyxin B (PB/50 µg/300UI)	S	1	1.72
	R	57	98.27

S: susceptible; R: resistant.

**Table 3 antibiotics-10-00846-t003:** Multidrug-resistant phenotype of *Pseudomonas aeruginosa* strains isolated from dogs with superficial infections: resistance (full circles) and susceptibility (empty circles).

Antimicrobial Agent	No. of Multidrug-Resistant *Pseudomonas aeruginosa* Tested Strains																	
	12	17	21	28	33	37	39	41	43	47	49	50	53	54	55	56	57	58
GM/10 µg	●	●	●	●	●	●	●	●	●	●	●	●	●	●	●	●	●	●
CIP/5 µg	●	●	●	●	●	●	●	●	●	●	●	●	●	●	●	●	●	●
IPM/10 µg	●	●	●	●	●	●	●	●	●	●	●	●	●	●	●	●	●	●
MEM/10 µg	●	●	●	●	●	●	●	●	●	●	●	●	●	●	●	●	●	●
TZP 110/100 + 10 µg	●	●	●	●	●	●	●	●	●	●	●	●	●	●	●	●	●	●
CAZ/30 µg	●	○	●	●	●	●	●	●	○	●	○	●	●	●	●	○	●	○
FEP/30 µg	●	●	●	●	●	●	○	●	●	●	●	●	●	●	●	●	●	○
ATM/30 µg	●	●	●	●	●	●	●	●	●	●	●	●	●	○	●	●	●	○
AZM/15 µg	●	●	●	●	○	●	●	●	○	●	●	●	●	●	●	●	●	●
AN 30 µg	○	○	●	○	●	●	●	●	●	●	●	●	○	●	●	●	●	●
TM/10 µg	●	●	●	●	●	●	●	●	●	●	●	●	●	●	●	●	●	●
PB/50 µg/300UI	●	●	●	●	●	●	●	●	●	●	●	●	●	●	●	●	●	●

**Table 4 antibiotics-10-00846-t004:** Results of antibiotic susceptibility testing of *Pseudomonas aeruginosa* isolates from the skin (*n* = 19).

I.D. of *Pseudomonas* Isolate	GM	CIP	IPM	MEM	TZP	CAZ	FEP	ATM	AZM	AN	TM	PB	No. of Resistant Antibiotics
3	S	R	R	R	R	S	S	S	R	S	R	R	7
7	R	R	R	R	R	S	S	S	S	S	R	R	8
8	S	R	S	S	R	R	R	S	S	S	R	R	7
9	R	S	R	S	R	S	R	S	R	S	R	R	8
14	R	R	S	R	R	S	S	S	R	S	R	R	7
17	R	R	R	R	R	S	R	R	R	S	R	R	11
20	R	R	R	S	R	S	R	S	S	R	R	R	8
21	R	R	R	R	R	R	R	R	R	R	R	R	13
32	R	R	R	R	R	S	S	S	S	R	R	R	8
36	S	S	S	S	S	S	S	S	S	S	R	R	2
39	R	R	R	R	R	R	S	R	R	R	R	R	11
40	R	R	S	R	S	R	S	R	S	S	R	R	8
43	R	R	R	R	R	S	R	R	S	R	R	R	11
49	R	R	R	R	R	S	R	R	R	R	R	R	12
50	R	R	R	R	R	R	R	R	R	R	R	R	13
51	R	S	S	R	S	R	R	S	S	R	S	R	7
52	S	S	R	S	R	S	R	R	R	S	R	R	8
53	R	R	R	R	R	R	R	R	R	S	R	R	12
58	R	R	R	R	R	S	S	S	R	R	R	R	10

S: susceptible; R: resistant.

**Table 5 antibiotics-10-00846-t005:** Results of antibiotic susceptibility testing of *Pseudomonas aeruginosa* isolates from otitis externa (*n* = 15).

I.D. of *Pseudomonas* Isolate	GM	CIP	IPM	MEM	TZP	CAZ	FEP	ATM	AZM	AN	TM	PB	No. of Resistant Antibiotics
1	S	R	R	R	R	S	S	S	R	S	R	R	7
2	R	S	R	S	R	S	S	S	S	R	R	R	7
6	S	R	R	R	R	R	R	S	S	S	R	R	9
12	R	R	R	R	R	R	R	R	R	S	R	R	12
13	R	R	R	R	R	S	S	S	R	S	R	R	8
15	R	R	S	S	R	S	R	R	R	S	R	R	8
16	R	S	S	R	S	S	R	S	R	R	R	R	7
23	S	R	R	R	R	S	S	S	R	R	R	R	8
25	R	R	S	S	S	S	S	S	S	R	R	R	5
28	R	R	R	R	R	R	R	R	R	S	R	R	12
29	S	R	R	R	R	S	S	S	S	R	R	R	7
30	R	S	S	R	S	R	S	R	R	R	R	R	8
35	R	S	S	S	R	S	R	R	S	R	R	R	8
56	R	R	R	R	R	S	R	R	R	R	R	R	12
57	R	R	R	R	R	R	R	R	R	R	R	R	13

S: susceptible; R: resistant.

**Table 6 antibiotics-10-00846-t006:** Results of antibiotic susceptibility testing of *Pseudomonas aeruginosa* isolates from perianal abscesses (*n* = 24).

I.D. of *Pseudomonas* Isolate	GM	CIP	IPM	MEM	TZP	CAZ	FEP	ATM	AZM	AN	TM	PB	No. of Resistant Antibiotics
4	S	R	R	R	S	S	S	S	S	S	R	R	5
5	R	S	S	R	R	R	S	S	S	R	R	R	8
10	R	R	R	R	R	S	R	S	S	S	R	R	8
11	S	R	R	S	S	S	R	S	R	R	R	R	7
18	S	R	R	R	R	R	R	S	S	R	R	S	8
19	S	R	R	S	S	S	S	S	R	R	R	R	6
22	S	R	R	R	S	S	S	S	S	R	R	R	6
24	S	R	R	R	R	S	S	S	R	R	R	R	8
26	S	R	R	R	S	R	R	R	S	S	R	R	7
27	R	S	S	R	S	R	R	R	R	S	R	R	8
31	R	R	S	S	R	S	R	R	S	S	R	R	7
33	R	R	R	R	R	R	R	R	S	R	R	R	12
34	S	R	R	S	R	R	R	S	S	R	R	R	8
37	R	R	R	R	R	R	R	R	R	R	R	R	13
38	S	R	R	R	R	S	R	S	R	S	S	R	7
41	R	R	R	R	R	R	R	R	R	R	R	R	13
42	S	R	R	R	S	S	R	R	R	S	S	R	7
44	S	R	R	R	R	S	S	R	R	S	S	R	7
45	S	R	R	R	R	R	R	S	S	S	R	R	8
46	S	R	R	S	S	R	R	S	S	R	R	R	7
47	R	R	R	R	R	R	R	R	R	R	R	R	13
48	S	R	R	S	S	R	R	R	R	S	S	R	7
54	R	R	R	R	R	R	R	S	R	R	R	R	12
55	R	R	R	R	R	R	R	R	R	R	R	R	13

S: susceptible; R: resistant.

**Table 7 antibiotics-10-00846-t007:** Primary criteria for the inclusion of dogs in the study, by the clinical score.

Clinical Score	Clinical Symptoms	Type of Diseases/No. of Collected Samples	No. of *P. aeruginosa* Strain Isolated
1	Excessive scaling, erythema	Superficial skin infection/*n* = 10	0
	Excessive scaling, erythema	Otitis externa/*n* = 8	0
	Excessive scaling, erythema	Perianal abscesses/*n* = 0	0
2	Presence of follicular papules, epidermal collarettes	Superficial skin infection/*n* = 15	2
		Otitis externa/*n* = 11	7
		Perianal abscesses/*n* = 0	0
3	Presence of follicular papules, epidermal collarettes, serous crusts, erythema, serohemorrhagic exudates, local pain	Superficial skin infection/*n* = 9	7
	Erythema, pruritus, ear discharge or desquamation, local pain	Otitis externa/*n* = 18	3
	swelled perianal glands, local pain	Perianal abscesses/*n* = 14	8
4	Presence of follicular papules, epidermal collarettes, serous crusts, erythema, serohemorrhagic or purulent exudates	Superficial skin infection/*n* = 22	10
	ear discharge with purulent exudates, intense local pain	Otitis externa/*n* = 11	5
	inflamed perianal glands, intense regional pain	Perianal abscesses/*n* = 24	16

## Data Availability

The datasets generated and analyzed during the current study are included within the article.

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
