# Peer review of "Antibiotic Susceptibility Profile of Pseudomonas aeruginosa Canine Isolates from a Multicentric Study in Romania"

_antibiotics, 2021, doi:10.3390/antibiotics10070846_

Round 1

Reviewer 1 Report

The manuscript 'Antibiotic susceptibility profile of Pseudomonas aeruginosa canine isolates from a multicentric study in Romania' by Degi et al studies the susceptibility or resistance of various drugs of canine infections. While I find the study interesting, there are some experiments described in the methods section but there are no results to support them.

Minor comments:

  1. a) Please correct the spelling of gentamycin to gentamicin throughout the manuscript. Similarly, hair follicle should be hair follicles.
  2. b) Instead of Eppendorf tubes, write microcentrifuge tubes
  3. c) At times, the manuscript becomes very difficult to follow. Please try to rewrite some sentences so that they become fluent. For example, Line 267 should be “particular importance” or “Particularly important”.

Major Comments:

  1. A) Add the results of the method section 3.3.1. The gel image needs to be shown.
  2. B) Rewrite the conclusions section. Try to incorporate all your findings and show their importance. A badly written conclusion does not make justice to the findings. There is not even a single reference in the conclusions section. Include the previously reported literature in your conclusions and show how your work contributes to the field of study.

Author Response

Dear Reviewer,

Many thanks for your evaluation and suggestions.

Answer to Reviewer 2

Dear Colleague many thanks for time spent to evaluate our work and to give us precious indications and suggestions. Our answers are in blue color.

Comments and Suggestions for Authors

The manuscript 'Antibiotic susceptibility profile of Pseudomonas aeruginosa canine isolates from a multicentric study in Romania' by Degi et al studies the susceptibility or resistance of various drugs of canine infections. While I find the study interesting, there are some experiments described in the methods section but there are no results to support them.

Minor comments:

  1. a) Please correct the spelling of gentamycin to gentamicin throughout the manuscript. Similarly, hair follicle should be hair follicles.

Answer: We have corrected in the manuscript.

  1. b) Instead of Eppendorf tubes, write microcentrifuge tubes

Answer: We have corrected in the manuscript

  1. c) At times, the manuscript becomes very difficult to follow. Please try to rewrite some sentences so that they become fluent. For example, Line 267 should be “particular importance” or “Particularly important”.

Answer: we rewritten all hard to read parts, to be more readable. Also we corrected to line 267.

Major Comments:

  1. A) Add the results of the method section 3.3.1. The gel image needs to be shown.

Answer: We have corrected and added as required. Many thanks for the suggestion.

  1. B) Rewrite the conclusions section. Try to incorporate all your findings and show their importance. A badly written conclusion does not make justice to the findings. There is not even a single reference in the conclusions section. Include the previously reported literature in your conclusions and show how your work contributes to the field of study.

Answer: Dear reviewer, from the “Author instructions” of Antibiotics, the citing references are to be included in the Discussions part. That is why you can find our findings commented and citations to this section.

Reviewer 2 Report

Antibiotic resistance is a growing global menace. In this study, Degi et al. report the antibiotic susceptibility of P. aeruginosa strains isolated from 142 dogs. This article would be of broad interest to readers of the antibiotics journal. However, some items need to be addressed before publication:

  1. My primary concern is the lack of genotype analysis for multi-resistant P. aeruginosa strains in Table 3. The authors can significantly strengthen their study by exploring which genes contribute to multi-drug resistance.
  2. The numbered list from line 229 to 247 should be corrected.
  3. It would be nice for authors to explain why use R* in Table 4 and R** in Table 5/6.

Overall Recommendation: Accept after minor revision (corrections to minor methodological errors and text editing)

Author Response

Dear Reviewer,

Many thanks for your evaluation and suggestions.

Answers to Reviewer 1.

Comments and Suggestions for Authors

Antibiotic resistance is a growing global menace. In this study, Degi et al. report the antibiotic susceptibility of P. aeruginosa strains isolated from 142 dogs. This article would be of broad interest to readers of the antibiotics journal. However, some items need to be addressed before publication:

  1. My primary concern is the lack of genotype analysis for multi-resistant P. aeruginosa strains in Table 3. The authors can significantly strengthen their study by exploring which genes contribute to multi-drug resistance.

Answer: The main goal of this study was to follow the clinical outcome of therapy (the stewardship part) and not especially the genetic of the Pseudomonas resistance, but currently we are studying also this feature of resistance. Even tough, as you observed in the manuscript, we had also a genetic investigation to identify Pseudomonas species, as positive control (confirming our ascertained strains).

  1. The numbered list from line 229 to 247 should be corrected.

Answer: We rewrote this part to be more understandable. Many thanks for the observation.

  1. It would be nice for authors to explain why use R* in Table 4 and R** in Table 5/6.

Answer: We harmonized the content for all manuscript. Many thanks for the observation.

Overall Recommendation: Accept after minor revision (corrections to minor methodological errors and text editing)

Answer: Many thanks for your time spent for our work evaluation and suggestions provided.

Round 2

Reviewer 1 Report

ACCEPTED AFTER REVISION

Reviewer 2 Report

The authors have addressed my concerns. The revised manuscript looks good.